# Relationship between Plasma Lipopolysaccharide Concentration and Health Status in Healthy Subjects and Patients with Abnormal Glucose Metabolism in Japan: A Preliminary Cross-Sectional Study

Nobuo Fuke [1],*, Shojiro Sawada [2],†, Takahiro Ito-Sasaki [3], Kumi Y. Inoue [3],‡, Yusuke Ushida [1], Ikuo Sato [4], Tomokazu Matsue [5], Hideki Katagiri [2], Hiroyuki Ueda [1] and Hiroyuki Suganuma [1]

[1] Innovation Division, KAGOME CO., LTD., 17 Nishitomiyama, Nasushiobara 329-2762, Tochigi, Japan
[2] Department of Metabolism and Diabetes, Tohoku University Graduate School of Medicine, 2-1 Seiryo-machi, Aoba-ku, Sendai 980-8575, Miyagi, Japan
[3] Graduate School of Environmental Studies, Tohoku University, 468-1 Aramaki Aza Aoba, Aoba-ku, Sendai 980-8572, Miyagi, Japan
[4] Department of Obstetrics and Gynecology, International University of Health and Welfare Hospital, 537-3 Iguchi, Nasushiobara 329-2762, Tochigi, Japan
[5] Center for Promotion of Innovation Strategy, Tohoku University, 468-1 Aramaki Aza Aoba, Aoba-ku, Sendai 980-8572, Miyagi, Japan
* Correspondence: nobuo_fuke@kagome.co.jp; Tel.: +81-80-1573-5815
† Current address: Division of Metabolism and Diabetes, Faculty of Medicine, Tohoku Medical and Pharmaceutical University, 1-15-1 Fukumuro, Miyagino-ku, Sendai 983-8536, Miyagi, Japan.
‡ Current address: Center for Basic Education, Faculty of Engineering, Graduate Faculty of Interdisciplinary Research, University of Yamanashi, 4-3-11 Takeda, Kofu 400-8511, Yamanashi, Japan.

**Abstract:** Lipopolysaccharides are components of Gram-negative bacteria. The relationship between blood lipopolysaccharide levels and health status has mainly been investigated in Europe, and there is a lack of information about Asia, particularly Japan. This study aimed to investigate the relationship between blood lipopolysaccharide levels and health status in the Japanese. We conducted two cross-sectional studies in 36 healthy subjects (Study 1) and 36 patients with abnormal glucose metabolism (AGM; Study 2). The plasma lipopolysaccharide concentration in healthy subjects was positively correlated with body mass index. The plasma lipopolysaccharide concentration in AGM patients was obviously higher than that in healthy subjects. Furthermore, in AGM patients, the plasma lipopolysaccharide concentration was positively correlated with C-peptide, fasting plasma glucose levels, triglycerides, and stage of diabetic nephropathy. The plasma lipopolysaccharide concentration was also negatively correlated with $20/(\text{C-peptide} \times \text{fasting plasma glucose})$, an indicator of insulin resistance, and high-density lipoprotein cholesterol. In particular, the correlation between plasma lipopolysaccharide concentration and triglycerides in AGM patients was maintained in multiple regression analyses adjusted for age, sex, or body mass index. These results suggest a possible role of lipopolysaccharides in obesity in healthy subjects and in the deterioration of triglyceride metabolism in AGM patients in the Japanese population.

**Keywords:** diabetes mellitus; lipopolysaccharides; metabolic endotoxemia

## 1. Introduction

Lipopolysaccharides (LPSs) are molecules in Gram-negative bacteria's outer membrane, made of lipids and polysaccharides. They are recognized endotoxins that activate toll-like receptor 4, causing inflammation in humans and animals. LPSs were thought to enter the human bloodstream only in conditions such as infectious disease and colitis. However, Cani et al. [1,2] found that high-fat diets raise gut-derived LPS influx, inflaming liver and adipose tissue, causing abnormalities in glucose metabolism and obesity. Cani et al. [1]

have defined this phenomenon as "metabolic endotoxemia". Subsequently, an association between obesity [3] or diabetes [4] and blood LPS levels was also observed in humans. The mechanisms by which circulating LPSs impair various metabolic functions have become increasingly elucidated. Specifically, when LPSs reach adipose tissue, they interact with adipose tissue-localised macrophages and induce the production of inflammatory cytokines, thereby inhibiting adipocyte browning [5]. As the browning of adipocytes plays a crucial role in the metabolisms of triglycerides [6], suppression of adipocyte browning by LPSs is thought to contribute to the decline in lipid metabolism, obesity, and increased blood pressure through arterial sclerosis. In addition, LPSs have been shown, in in vitro studies using myoblast cell lines, to increase insulin resistance through inducible nitric oxide synthase-mediated tyrosine nitration of insulin receptor substrate 1 [7]. Furthermore, LPSs have been reported to induce apoptosis in hepatocytes through toll-like receptor 4, leading to liver damage [8]. Consequently, it is hypothesised that circulating LPSs impair cellular functions in various organs, leading to a deterioration in metabolic capabilities. Thus, intervention studies targeting the blood LPS level as a therapeutic target have been conducted in patients with obesity, metabolic syndrome, and type 2 diabetes [9]. Against this background, blood LPS level is expected to be a promising biomarker for preventing and treating lifestyle-related diseases in the future.

Gnauck et al. reported that blood LPS concentration varied widely, ranging from under the detection limit (<0.005) to 61 EU/mL in healthy individuals, from 56 to 67 EU/mL in individuals with type 1 diabetes, and from under the detection limit (<0.005) to 77 EU/mL in individuals with type 2 diabetes [10]. The reasons for this variation may be differences in lifestyle, living environment, race, gut microbiota, and measuring methods. To utilize blood LPS level as an indicator for health management, it is important to assess blood LPSs using a uniform and appropriate measurement method and to comprehend the distribution of LPS levels and their relationship to health status in countries and regions intending to employ this approach. However, the relationship between blood LPS levels and health status has mainly been investigated in Europe. In the Japanese, only a few previous studies have assessed the association between obesity/diabetes and blood LPSs. Specifically, the blood levels of LPS-binding protein (LBP) have been reported to be associated with insulin sensitivity [11] and the risk of developing metabolic syndrome in the future [12] in studies among healthy subjects. It has also been reported that patients with type 2 diabetes have higher blood concentrations of LBP than healthy subjects [13] and that blood concentrations of LBP correlate with body mass index (BMI), glucose metabolism, lipid metabolism, and inflammatory indices in diabetic patients [14]. Blood levels of LBP have also been reported to be associated with BMI and inflammatory indices in patients with type 1 diabetes [15]. However, all of these previous studies in the Japanese population have evaluated LBP rather than LPSs. LBP is a protein produced mainly in the liver in response to LPSs [16]. Therefore, blood LBP is utilized as a surrogate marker for blood LPSs in human studies [17–19]. However, since a certain amount of LBP is produced in the liver even in the absence of overt infection [20], since the production of LBP in the liver is reduced in some liver diseases (e.g., liver injury) [21], and since LBP is also produced by differentiated adipocytes in an LPS-independent manner [22], it can be assumed that factors other than LPS influx may cause changes in blood LBP levels. Indeed, blood LPS and LBP concentrations do not always coincide [23]. Therefore, epidemiological studies measuring the LPS level itself are needed to clarify the relationship between blood LPSs and obesity/diabetes in the Japanese population.

A previous report measuring blood LBP levels in Japanese diabetic patients pointed to the short half-life of blood LPSs as the reason for not measuring blood LPSs [14]. In a previous report measuring blood LBP levels in healthy Japanese subjects and patients with type 2 diabetes, blood LPSs were also measured but not detected [13]. Indeed, blood LPSs are unstable, and their concentration decreases even if samples are stored at $-80$ °C [24]. We, therefore, attempted to complete the process from plasma isolation to measurement of LPS concentration on "the day of blood collection" in a preliminary study in four

healthy Japanese subjects. As a result, LPSs were detected in plasma; furthermore, their concentration was found to be associated with appetite [25].

Therefore, in the present study, plasma LPS concentrations were measured in healthy subjects (Study 1) and in patients with abnormal glucose metabolism (Study 2), consisting of diabetes mellitus and impaired glucose tolerance (IGT) subjects, in the same way as previously reported [25], and their association with the health status of the study subjects was assessed. Note that there are two types of diabetes: type 2 diabetes, in which obesity contributes to the onset of the disease, and type 1 diabetes, in which an autoimmune response is the main cause. Type 2 diabetes is frequently reported to be associated with blood LPSs, but an increase in LPS-producing bacteria in the gut microbiota [26,27], an increase in blood LPS levels [28], and an association between blood LPS levels and the severity of complications [29,30] have also been reported in type 1 diabetes. Therefore, the study included patients with both type 1 and type 2 diabetes. Data analyses were then performed even when patients with type 1 diabetes were excluded.

## 2. Materials and Methods

### 2.1. Ethical Approval and Consent to Participate

This study was conducted in accordance with the guidelines of the 2013 version of the Declaration of Helsinki, and all procedures involving human subjects were approved by the ethics committees of KAGOME CO., LTD. (Tokyo, Japan) (approval Nos. 2017-R03 for Study 1 and 2017-R13 for Study 2), and the ethics committees of Tohoku University (approval No. 2018-1-061 for Study 2). Written informed consent was obtained from all participants.

Although the participants in Study 1 were employees at the same company as the researchers, during the study briefing, it was explained, both verbally and in writing, that participation or withdrawal from the study was voluntary and that there were no disadvantages to not participating or withdrawing from the study.

### 2.2. Study Design

Studies 1 and 2 were cross-sectional studies, conducted from March 2017 to December 2017 in Tochigi, Japan, and from December 2017 to June 2018 in Miyagi, Japan, respectively.

Given the involvement of a private company in this study, the potential for bias in the study results was addressed; in Study 1, the association between plasma LPS concentrations and demographic data was evaluated. Therefore, before measuring plasma LPS concentrations, a medical interview was conducted to avoid any bias when subjects completed the questionnaire. Furthermore, to avoid any bias when the researchers measured the plasma LPS concentrations, the individuals' demographic data were disclosed to the researchers after the measurement of the plasma LPS concentrations was completed. This eliminated the possibility of bias in the data. In Study 2, researchers from KAGOME CO., LTD., were involved in the development of the study protocol but not in the collection of data, thereby eliminating the possibility of bias in the study results.

### 2.3. Study 1

2.3.1. Subjects

Male and female KAGOME CO., LTD., employees aged ≥20 years were recruited via email. Individuals who did not meet the following exclusion criteria were included: (1) subjects who could not be disinfected with ethanol due to ethanol sensitivity; (2) subjects who were breastfeeding, pregnant, or wanted to become pregnant during the trial period; (3) subjects who had any concerns about their condition after blood collection.

2.3.2. Schedule

One month before blood collection, the subjects were interviewed by the study physician. Subjects fasted overnight (>12 h) from 21:00 on the day before blood collection, and the blood was drawn between 9:00 am and 12:00 pm on the next day. The blood samples

were subjected to measurements of the concentration of LPSs on the same day on which the blood samples were taken.

### 2.3.3. Data Collection

1. Medical interview

Previous reports have suggested that age [31], sex [31], BMI [31], smoking [32], alcohol consumption [31], energy intake [32], and lipid intake [32] are associated with blood LPS levels. Therefore, information on these items was obtained in the medical interview. Specifically, subjects completed a questionnaire on sex, age, BMI, whether they had metabolic syndrome at their health examination within a year, medical history, current medical history, medication use, smoking habits, drinking habits, and dietary habits. With regard to dietary habits, the subjects themselves answered the following questions for the purpose of a simplified assessment of energy and fat intake: number of meals per day (2 or less, or 3 times), snacking frequency per week (infrequent, several times per week, or daily), and frequency of fatty meals per week (infrequent, several times per week, or daily).

2. Blood samples

After disinfection of the blood collection site with ethanol, 5 mL of blood was collected. Blood samples were stored on ice immediately after collection. Blood samples were then centrifuged at $1200 \times g$ at 4 °C for 15 min, and the supernatant (heparin plasma) was collected.

3. Analysis of plasma LPS concentrations

The measurement of plasma LPS concentrations was performed as previously reported [25]. In brief, plasma samples were diluted 10-fold with LPS-free distilled water (Otsuka Pharmaceutical Co., Ltd., Tokyo, Japan), then heated to 70 °C for 10 min, and sonicated at 37 °C for 10 min. Then, 50 μL of each sample and limulus amoebocyte lysate (LAL) reagent (Endospecy; SEIKAGAKU CORPORATION, Tokyo, Japan) were mixed in a 96-well plate (AGC TECHNO GLASS Co., Ltd., Shizuoka, Japan) and incubated at 37 °C for 60 min (plate #1). To correct the absorbance (Abs) originating from the colour of the plasma, 50 μL of the plasma and LPS-free distilled water were mixed in a 96-well plate and incubated at 37 °C for 60 min (plate #2). After incubation, the Abs was measured at 405 nm (reference Abs: 492 nm) using a microplate reader (CORONA ELECTRIC Co., Ltd., Ibaraki, Japan). The Abs originating from the LAL reaction was calculated as follows:

(Abs of 'plasma + LAL reagent' well [plate #1]) − (Abs of blank well [plate #1]) − ([Abs of 'plasma + water' well {plate #2}] − [Abs of blank well {plate #2}])

Control standard endotoxin (SEIKAGAKU CORPORATION, Chiyoda-ku, Tokyo) in the range of 0.0001–0.1000 EU/mL was reacted with LAL reagent in the same way as the plasma in plate #1. A calibration curve was drawn with control standard endotoxin (SEIKAGAKU CORPORATION) based on the LPS concentration in the plasma sample. Plasma samples under the detection limit (0.0001 EU/mL) were assumed to have 0 EU/mL LPSs in the data analysis.

### 2.4. Study 2

#### 2.4.1. Subjects

Patients aged ≥20 years who were admitted to Tohoku University Hospital for AGM and did not meet the following exclusion criteria were included: (1) patients in the acute phase of an infectious disease; (2) patients with gastrointestinal disorders that may lead to a high blood LPS concentration. In this study, diabetes mellitus and IGT were defined as AGM. Diabetes mellitus and IGT were defined in accordance with American Diabetes Association guidelines [33]. For patients with diabetes, the discharge criteria were fasting glucose level <130 mg/dL and 2 h post-load plasma glucose level <180 mg/dL.

### 2.4.2. Schedule

Patients were interviewed on the first day of hospitalization. Physical measurements and blood collections were performed on the second day of admission (referred to as "at admission" hereafter) and the day before discharge (referred to as "at discharge" hereafter). To observe the effect of hospitalization on metabolic parameters, only patients who were hospitalized for ≥7 days were included. During hospitalization, patients were given lifestyle guidance, such as diet and exercise guidance, in accordance with Japan Diabetes Society guidelines. Specifically, their diet was calculated according to 30 kcal × ideal body weight (=22 × height [m$^2$]) (kg). We also recommended that subjects walked 10,000 steps per day.

### 2.4.3. Data Collection

1. Medical interview

   The doctors interviewed all patients regarding their sex, age, and medical history.

2. Blood samples and analysis of plasma LPS concentration

   The blood samples were collected and analysed using the methods described in Study 1.

3. Physiological and biochemical analyses

   Height, weight, systolic blood pressure (SBP), levels of fasting plasma glucose (FPG), glycated albumin (GA), haemoglobin A1c (HbA1c), C-peptide (CPR), triglycerides (TGs), total cholesterol (TC), low-density lipoprotein cholesterol (LDL-C), high-density lipoprotein cholesterol (HDL-C), high-sensitivity C-reactive protein (hs-CRP), liver function markers (alanine aminotransferase (ALT), aspartate aminotransferase (AST), gamma-glutamyl transferase (γ-GTP)), urine albumin, and estimated glomerular filtration rate (eGFR) were measured according to standard clinical tests in hospitals. The stage of diabetic nephropathy was assessed using urine albumin and eGFR, as previously described [34]. BMI was calculated based on patient height and weight. Insulin resistance was calculated as follows, as previously described [35]:

$$20/(CPR \times FPG)$$

### 2.5. Statistical Methods

The sample size for Study 1 was determined based on previous reports [36–38]. Previous cross-sectional studies conducted to evaluate the correlation between blood LPS concentration and health status in Italy [36] and Finland [37] had 24 and 25 subjects, respectively. Furthermore, based on previous reports evaluating the correlation analysis between blood LPS concentration and BMI [38], as in Study 1, the sample size was calculated assuming a correlation coefficient between plasma LPS concentration and BMI of 0.46 (the highest correlation coefficient in the paper), a significance level of 5% (two-sided), and power of 80%. As a result, the sample size required for Study 1 was set to ≥32. The sample size for Study 2 was determined based on a previous report [39,40]. Specifically, in a previous study, serum LPS concentrations of patients with diabetes were considered their plasma LPS concentrations "at admission", while serum LPS concentrations in healthy subjects were considered their plasma LPS concentrations "at discharge" [39]. The difference between the concentrations "at admission" and "at discharge" was assumed to be 0.1, and the standard deviation (SD) of each group was assumed to be 0.1. When a significance level of 5% (two-sided) and power of 80% were set, the number of patients with AGM required was 17. Furthermore, based on a previous report evaluating the correlation between blood LPS levels and clinical parameters in diabetic patients [40], a correlation coefficient of 0.477 (the highest correlation coefficient in the paper) was assumed, and the sample size was calculated using a significance level of 5% (two-sided) and power of 80%. As a result, the sample size required for Study 2 was set to ≥29.

Data are presented as medians and interquartile ranges (IQRs) unless otherwise stated. The normality of all data was tested using the Shapiro–Wilk normality test. The paired *t*-test was used to compare the means of the two groups with normally distributed data. The Wilcoxon signed-rank test was used to compare the median between two paired groups of non-normally distributed data. The Mann–Whitney *U* test was used to compare the median between two unpaired groups. For correlation analysis, Spearman rank correlation coefficients were calculated, because plasma LPS concentrations were not normally distributed (Table S1). Fisher's exact test was used to compare sex ratios between groups. When comparing plasma LPS concentrations between healthy subjects and AGM ones, analysis of covariance (ANCOVA) was used, with plasma LPS concentration as the objective variable and age or BMI as a correction factor, in order to eliminate the influence of confounding factors. In the multiple regression analysis, physiological and biochemical test values were adopted as objective variables; plasma LPS concentration was adopted as the explanatory variable; and age, sex, BMI, TC, LDL-C, HDL-C, TGs, 20/(CPR × FPG), stage of diabetic nephropathy, and insulin use were adopted as adjustment factors. In this study, the *p*-value threshold was adjusted using the Benjamini–Hochberg method, as the statistical analysis was repeated on the same dataset. In general, the false discovery rate (FDR) threshold is chosen among 0.05 to detect strong results [41], 0.1 to detect promising results [41], and 0.2 to detect possible results without missing any possible results while allowing for the possibility of false positives [42]. As this was a preliminary study aimed at investigating whether plasma LPS concentrations are associated with health status in the Japanese population, the FDR threshold was set to *Q* = 0.2 to avoid missing demographic data and clinical parameters that are associated with plasma LPS concentrations. The specific statistical methods used are described in the footnotes of the tables and figure legends. All statistical analysis were two-sided and performed using EZR ver. 1.40 [43].

## 3. Results

### 3.1. Study 1

#### 3.1.1. Subjects' Characteristics

Thirty-six subjects participated in Study 1 (Figure 1). The results of the Shapiro–Wilk test for each test value is shown in Table S1. As the result of the medical interview, none of the subjects had history of metabolic syndrome, dyslipidaemia, hypertension, or diabetes mellitus at their health examination within a year (Table 1); thus, we considered them "healthy subjects". One of the study subjects had a history of gastric ulcer; three had a history of appendicitis; and one had a history of peritonitis. One of them had a history of both appendicitis and peritonitis. There were no study subjects with suspected acute infections.

**Table 1.** Characteristics of healthy subjects (Study 1).

| Characteristic | Number or Median (IQR) | |
|---|---|---|
| *n* | 36 | |
| Age (years) | 33 | (29–39) |
| Sex (male:female) | 24:12 | |
| Body mass index (kg/m$^2$) | 21 | (20–23) |
| Diabetes (yes/no) | 0/36 | |
| Metabolic syndrome (yes/no) | 0/36 | |
| Habitual drinking (yes/no) | 25/11 | |
| Current smoker (yes/no) | 4/32 | |

IQR, interquartile range.

#### 3.1.2. Plasma LPS Concentrations in Healthy Japanese Subjects

The median LPS concentration was 0.0036 EU/mL, with a range of 0–0.0260 EU/mL (minimum–maximum; Figure 2). The plasma LPS concentration was significantly correlated with BMI (*p* = 0.02, *Q* = 0.09; Figure 3A) but not with age, sex, drinking habits, and smoking

habits (Figure 3B–E). The plasma LPS concentration was not associated with the number of meals per day, snacking frequency per week, or frequency of fatty meals per week (Figures S1–S3). Plasma LPS levels were compared between those with and without a history of gastrointestinal diseases, but no significant differences were found (Figure S4).

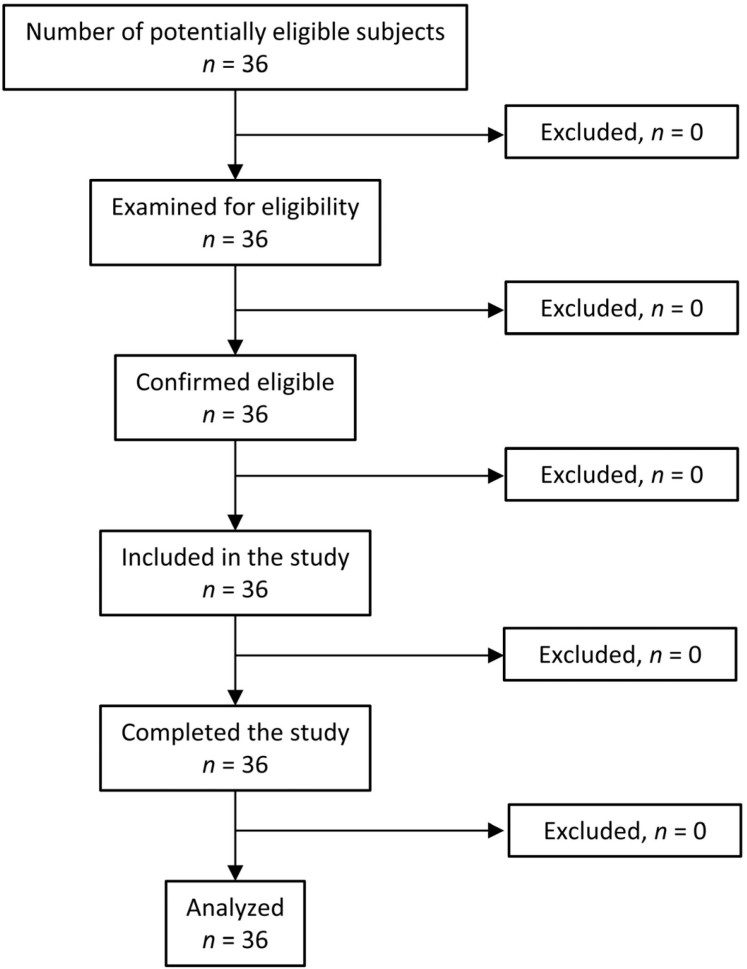

**Figure 1.** Flow diagram describing participant selection in Study 1.

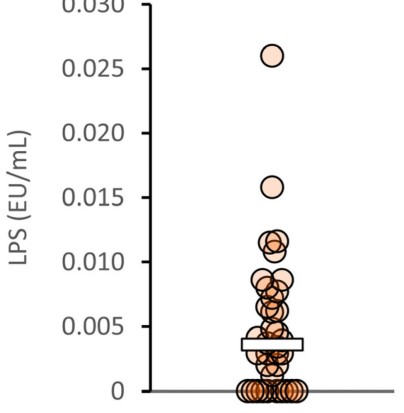

**Figure 2.** Distribution of plasma LPS concentrations in healthy subjects. Plasma samples were collected from healthy subjects, and LPS concentrations were measured on the same day. Circles represent individual values, and horizontal bars represent median values. *n* = 36. LPS, lipopolysaccharide.

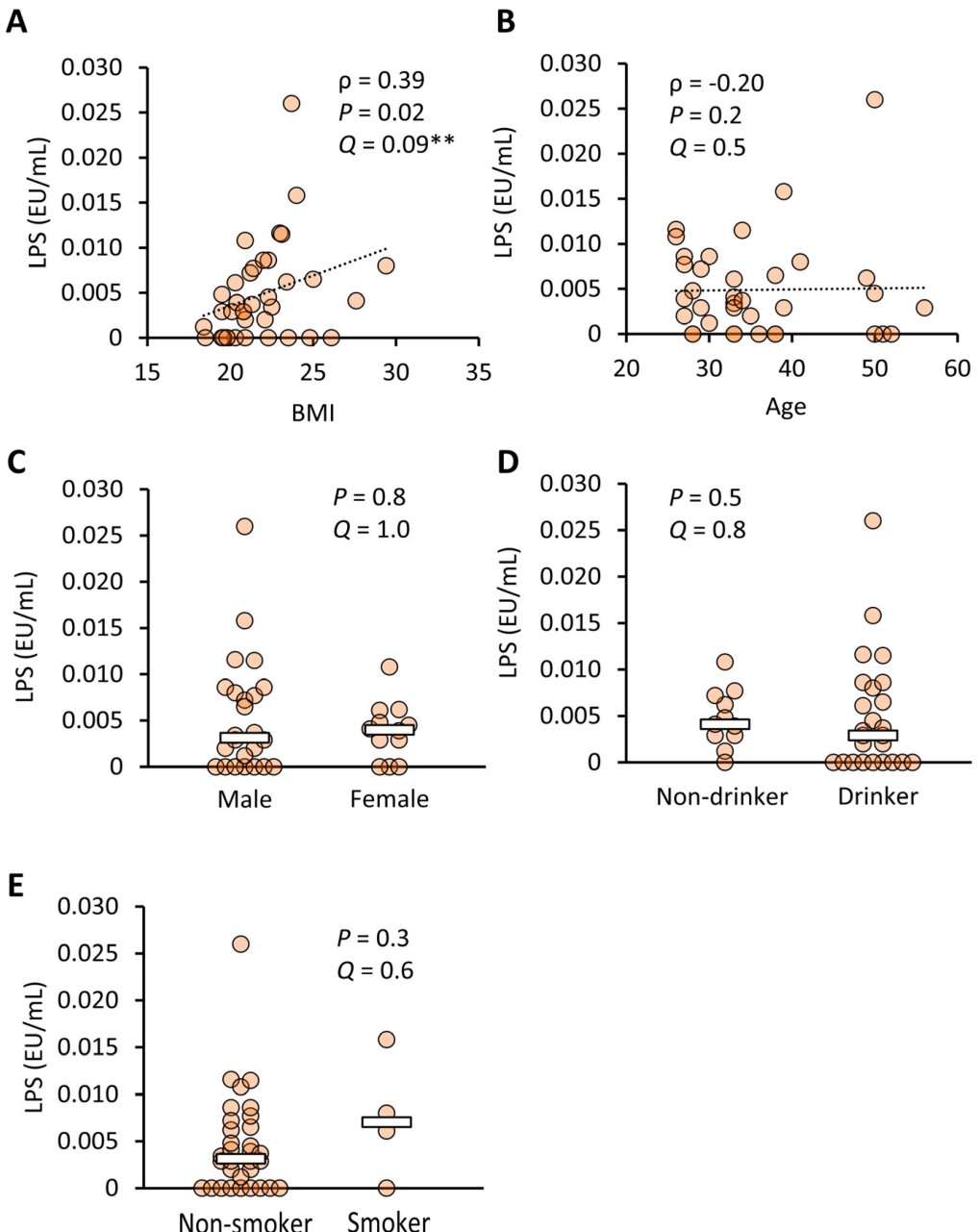

**Figure 3.** Relationships between plasma LPS concentrations and subject characteristics. (**A,B**) Correlation analyses between plasma LPS concentrations and BMI (**A**) or age (**B**). Circles represent individual values; ρ, *p*, and *Q* represent Spearman rank correlation coefficients, *p*-values, and *Q*-values, respectively (*n* = 36). ** *Q* < 0.1. (**C–E**) Plasma LPS concentrations stratified by sex (**C**), drinking habits (**D**), or smoking habits (**E**). Circles represent individual values, and horizontal bars represent median values. Comparisons of plasma LPS concentrations between groups were made using the Mann–Whitney *U* test, because plasma LPS concentrations were not normally distributed (Table S1). However, none of the differences were significant (*Q* < 0.2). For n in each stratum, see Table 1. LPS, lipopolysaccharide; BMI, body mass index.

*3.2. Study 2*

3.2.1. Subjects' Characteristics

Forty subjects were included in Study 2, but four subjects dropped out due to an early discharge because of faster improvements in blood glucose levels (Figure 4). Thirty-six subjects were included in the final analysis. The characteristics of the subjects in Study 2

are listed in Table 2. The results of the Shapiro–Wilk test for each test value is shown in Table S1.

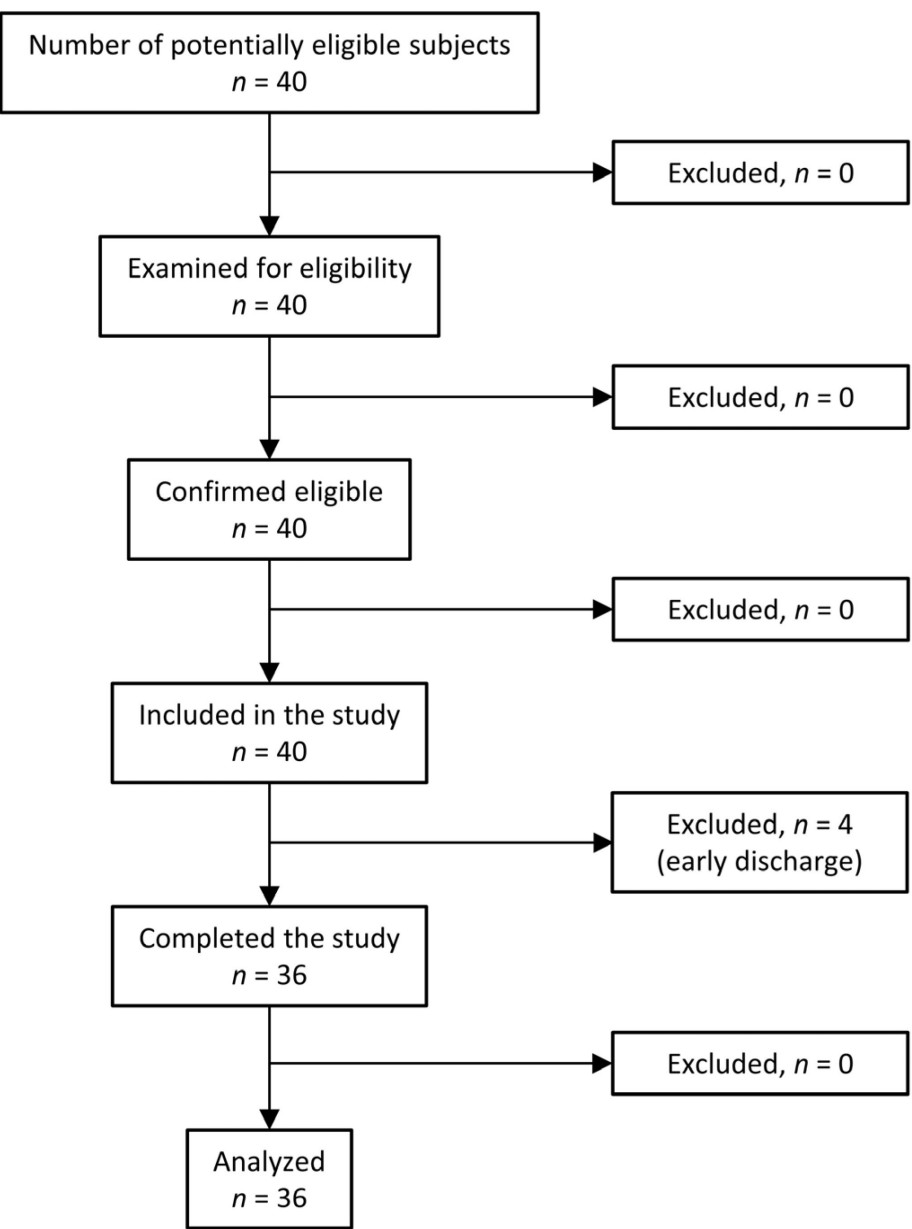

**Figure 4.** Flow diagram describing participant selection in Study 2.

3.2.2. Plasma LPS Concentrations in Japanese Patients with AGM

The median LPS concentration at admission was 0.0232 EU/mL, with a range of 0.0048–0.1205 EU/mL (minimum–maximum). There was no significant difference in plasma LPS concentrations at admission and discharge (Table 2).

Basic characteristics, namely, age, sex, and BMI, were compared between healthy subjects and AGM patients. The results of the Mann–Whitney $U$ test for age and BMI showed significant between-group differences ($p = 1.6 \times 10^{-9}$, $Q = 8.6 \times 10^{-8}$; $p = 3.6 \times 10^{-6}$, $Q = 4.4 \times 10^{-5}$, respectively). However, no significant difference was observed for sex using Fisher's exact probability test ($p = 0.6$, $Q = 0.9$). ANCOVA with age or BMI as an adjustment factor was then used to compare plasma LPS concentrations between groups. Regardless of the adjustment factor, the plasma LPS concentration was significantly higher

in AGM patients compared with healthy subjects ($p = 1.3 \times 10^{-5}$, $Q = 0.0002$; $p = 2.7 \times 10^{-7}$, $Q = 4.8 \times 10^{-6}$, respectively).

**Table 2.** Characteristics of patients with AGM (Study 2).

| Characteristic | At Admission | At Discharge | $p$ | $Q$ |
|---|---|---|---|---|
| | Number or Median (IQR) | Number or Median (IQR) | | |
| *n* | 36 | - | - | - |
| Age (years) | 64 (52–73) | - | - | - |
| Sex (male:female) | 21:15 | - | - | - |
| Diabetes (*n*) | | | | |
| Type 1 | 3 | - | - | - |
| Type 2 | 30 | - | - | - |
| IGT | 3 | - | - | - |
| Insulin use | 28 | - | - | - |
| CPR (ng/mL) | 2.1 (1.5–3.3) † | - | - | - |
| HbA1c (%) | 8.7 (6.8–10.0) | - | - | - |
| Stage of diabetic nephropathy (1:2:3:4:5:ND) | 22:3:8:0:1:2 | - | - | - |
| 20/(CPR × FPG) | 1.3 (0.9–1.8) † | - | - | - |
| FPG (mg/dL) | 130 (96–151) | 103 (88–119) | <0.01 | <0.05 *** |
| GA (%) | 22 (18–27) | 20 (15–22) | <0.01 | <0.05 *** |
| TGs (mg/dL) | 109 (85–128) | 109 (87–126) | 0.17 | 0.37 |
| TC (mg/dL) | 174 (162–199) | 156 (131–179) | <0.01 | <0.05 *** |
| HDL-C (mg/dL) | 40 (36–48) | 40 (37–47) | 0.15 | 0.36 |
| LDL-C (mg/dL) | 105 (87–121) | 90 (69–116) | <0.01 | <0.05 *** |
| AST (U/L) | 21 (18–28) | 22 (19–30) | 0.19 | 0.39 |
| ALT (U/L) | 22 (18–31) | 25 (16–31) | 0.59 | 0.83 |
| γ-GTP (U/L) | 29 (16–43) | 23 (14–45) | <0.01 | <0.05 *** |
| SBP (mmHg) | 124 (113–144) | 119 (111–125) | <0.01 | <0.05 *** |
| BMI (kg/m$^2$) | 26 (23–31) | 25 (23–30) | <0.01 | <0.05 *** |
| hs-CRP (mg/dL) | 0.101 (0.037–0.193) | 0.076 (0.028–0.184) | 0.1 | 0.30 |
| LPSs (EU/mL) | 0.0232 (0.0187–0.0306) | 0.0211 (0.0113–0.0286) | 0.12 | 0.35 |

AGM, abnormal glucose metabolism; IQR, interquartile range; IGT, impaired glucose tolerance; CPR, C-peptide; FPG, fasting plasma glucose; GA, glycated albumin; TGs, triglycerides; TC, total cholesterol; HDL-C, high-density lipoprotein cholesterol; LDL-C, low-density lipoprotein cholesterol; AST, aspartate aminotransferase; ALT, alanine aminotransferase; γ-GTP, γ-glutamyl transferase; SBP, systolic blood pressure; BMI, body mass index; hs-CRP, high-sensitivity C-reactive protein; HbA1c, haemoglobin A1c; ND, not determined; LPSs, lipopolysaccharides. Data are represented as numbers or medians (IQRs). *** $Q < 0.05$. As TC and LDL-C were normally distributed (Table S1), the paired-*t* test was used for these analyses. The other items were not normally distributed (Table S1), so the Wilcoxon signed-rank test was used. † Type 1 diabetes was excluded because insulin secretion was reduced due to an autoimmune response.

At admission, the plasma LPS concentration showed a significant positive correlation with CPR, FPG, TG levels, and stage of diabetic nephropathy and showed a significant negative correlation with 20/(CPR × FPG), independently of age, sex, or BMI (Table 3). As indicated in the previous reports [44,45], blood CPR levels in insulin users were significantly lower than those in non-users (Figure S5; $p = 0.005$, $Q = 0.03$). However, the significant correlation between the plasma LPS concentration and 20/(CPR × FPG) remained after adjusting for insulin use ($p = 0.01$, $Q = 0.07$). Significant correlations between the plasma LPS concentration and stage of diabetic nephropathy or 20/(CPR × FPG) disappeared after adjusting for TG levels ($p = 0.16$, $Q = 0.4$; $p = 0.52$, $Q = 0.8$, respectively). In contrast, a significant correlation between the plasma LPS concentration and TG levels was independent of stage of diabetic nephropathy ($p < 0.01$) or 20/(CPR × FPG) ($p = 1.5 \times 10^{-6}$, $Q = 2.2 \times 10^{-5}$; $p = 3.4 \times 10^{-6}$, $Q = 4.3 \times 10^{-5}$, respectively).

At discharge, the plasma LPS concentration showed a significant positive correlation with TG levels and showed a significant negative correlation with HDL-C levels, independently of age, sex, or BMI (Table 4). The significant correlation between plasma LPS concentration and HDL-C disappeared after adjusting for TGs ($p = 0.68$, $Q = 0.9$). In contrast, a significant correlation between the plasma LPS concentration and TG levels remained after adjusting for HDL-C levels ($p = 1.4 \times 10^{-8}$, $Q = 4.4 \times 10^{-7}$).

**Table 3.** Multiple regression analysis between physiological and biochemical test values and the plasma LPS concentration in patients at admission.

| Characteristic | Model 1 | | | Model 2 | | | Model 3 | | | Model 4 | | |
|---|---|---|---|---|---|---|---|---|---|---|---|---|
| | β | *p* | *Q* | β | *p* | *Q* | β | *p* | *Q* | β | *p* | *Q* |
| CPR | 42 | <0.01 | <0.05 *** | 43 | <0.01 | <0.05 *** | 42 | <0.01 | <0.05 *** | 42 | <0.01 | <0.05 *** |
| HbA1c | 34 | 0.15 | 0.36 | 34 | 0.15 | 0.36 | 34 | 0.15 | 0.36 | 36 | 0.07 | 0.22 |
| Diabetic nephropathy | 20 | <0.05 | <0.1 ** | 21 | <0.05 | <0.1 ** | 21 | <0.05 | <0.1 ** | 22 | <0.01 | <0.1 ** |
| 20/(CPR × FPG) | −24 | <0.05 | <0.2 * | −25 | <0.05 | <0.1 ** | −24 | <0.05 | <0.2 * | −24 | <0.05 | <0.2 * |
| FPG | 801 | <0.05 | <0.2 * | 800 | <0.05 | <0.2 * | 790 | <0.05 | <0.2 * | 827 | <0.05 | <0.2 * |
| GA | −0.7 | 0.99 | 1.00 | 0 | 1.00 | 1.00 | 0 | 1.00 | 1.00 | 12 | 0.84 | 0.97 |
| TGs | 2910 | <0.01 | <0.05 *** | 2910 | <0.01 | <0.05 *** | 2934 | <0.01 | <0.05 *** | 2904 | <0.01 | <0.05 *** |
| TC | 185 | 0.53 | 0.79 | 185 | 0.54 | 0.79 | 226 | 0.35 | 0.59 | 180 | 0.55 | 0.79 |
| LDL-C | 332 | <0.05 | 0.83 | 185 | 0.54 | 0.79 | 226 | 0.35 | 0.59 | 180 | 0.55 | 0.79 |
| HDL-C | −171 | 0.10 | 0.31 | −170 | 0.11 | 0.31 | −163 | 0.11 | 0.31 | −165 | 0.11 | 0.32 |
| AST | −207 | 0.16 | 0.36 | −209 | 0.13 | 0.36 | −209 | 0.16 | 0.36 | −212 | 0.15 | 0.36 |
| ALT | −350 | 0.35 | 0.59 | −358 | 0.28 | 0.51 | −356 | 0.35 | 0.59 | −365 | 0.33 | 0.58 |
| γ-GTP | −485 | 0.23 | 0.46 | −486 | 0.24 | 0.46 | −507 | 0.21 | 0.43 | −486 | 0.24 | 0.46 |
| SBP | −1.7 | 0.99 | 1.00 | 0 | 1.00 | 1.00 | 10 | 0.96 | 1.00 | 0 | 1.00 | 1.00 |
| BMI | 16 | 0.83 | 0.97 | 14 | 0.82 | 0.97 | 15 | 0.84 | 0.97 | - | - | - |
| hs-CRP | −0.4 | 0.85 | 0.97 | 0 | 0.84 | 0.97 | 0 | 0.88 | 0.97 | −1 | 0.67 | 0.90 |

LPSs, lipopolysaccharides; CPR, C-peptide; FPG, fasting plasma glucose; GA, glycated albumin; TGs, triglycerides; TC, total cholesterol; HDL-C, high-density lipoprotein cholesterol; LDL-C, low-density lipoprotein cholesterol; AST, aspartate aminotransferase; ALT, alanine aminotransferase; γ-GTP, γ-glutamyl transferase; SBP, systolic blood pressure; BMI, body mass index; hs-CRP, high-sensitivity C-reactive protein; HbA1c, haemoglobin A1c. * *Q* < 0.2, ** *Q* < 0.1, or *** *Q* < 0.05. Multiple regression analysis was performed. Model 1: unadjusted; Model 2: adjusted for age; Model 3: adjusted for sex; Model 4: adjusted for BMI.

**Table 4.** Multiple regression analysis between physiological and biochemical test values and the plasma LPS concentration in patients at discharge.

| Characteristic | Model 1 | | | Model 2 | | | Model 3 | | | Model 4 | | |
|---|---|---|---|---|---|---|---|---|---|---|---|---|
| | β | *p* | *Q* | β | *p* | *Q* | β | *p* | *Q* | β | *p* | *Q* |
| FPG | 94 | 0.64 | 0.88 | 113 | 0.54 | 0.79 | 79 | 0.69 | 0.90 | 64 | 0.72 | 0.94 |
| GA | 3 | 0.97 | 1.00 | 8 | 0.89 | 0.98 | 2 | 0.97 | 1.00 | −11 | 0.80 | 0.97 |
| TGs | 2434 | <0.01 | <0.05 *** | 2431 | <0.01 | <0.05 *** | 2416 | <0.01 | <0.05 *** | 2450 | <0.01 | <0.05 *** |
| TC | −274 | 0.37 | 0.61 | −269 | 0.38 | 0.62 | −301 | 0.29 | 0.53 | −269 | 0.39 | 0.62 |
| LDL-C | −421 | 0.12 | 0.35 | −426 | 0.12 | 0.35 | −438 | 0.10 | 0.30 | −402 | 0.14 | 0.36 |
| HDL-C | −168 | <0.05 | <0.2 * | −162 | <0.05 | <0.2 * | −172 | <0.05 | <0.2 * | −180 | <0.05 | <0.1 ** |
| AST | −400 | 0.15 | 0.36 | −402 | 0.15 | 0.36 | −386 | 0.15 | 0.36 | −415 | 0.13 | 0.36 |
| ALT | −604 | 0.14 | 0.36 | −624 | 0.13 | 0.35 | −583 | 0.15 | 0.36 | −616 | 0.14 | 0.36 |
| γ-GTP | −449 | 0.22 | 0.44 | −451 | 0.22 | 0.45 | −430 | 0.23 | 0.46 | −455 | 0.22 | 0.44 |
| SBP | 20 | 0.85 | 0.97 | 22 | 0.84 | 0.97 | 18 | 0.87 | 0.97 | 21 | 0.84 | 0.97 |
| BMI | −20 | 0.76 | 0.97 | −27 | 0.65 | 0.88 | −20 | 0.77 | 0.97 | - | - | - |
| hs-CRP | −1 | 0.63 | 0.87 | −1 | 0.60 | 0.85 | −1 | 0.64 | 0.87 | 0 | 0.71 | 0.93 |

LPSs, lipopolysaccharides; FPG, fasting plasma glucose; GA, glycated albumin; TGs, triglycerides; TC, total cholesterol; HDL-C, high-density lipoprotein cholesterol; LDL-C, low-density lipoprotein cholesterol; AST, aspartate aminotransferase; ALT, alanine aminotransferase; γ-GTP, γ-glutamyl transferase; SBP, systolic blood pressure; BMI, body mass index; hs-CRP, high-sensitivity C-reactive protein. * *Q* < 0.2, ** *Q* < 0.1, or *** *Q* < 0.05. Multiple regression analysis was performed. Model 1: unadjusted; Model 2: adjusted for age; Model 3: adjusted for sex; Model 4: adjusted for BMI.

Changes in the plasma LPS concentration from admission to discharge showed a significant positive correlation with changes in TG levels and showed a significant negative correlation with changes in HDL-C levels independently of changes in BMI (Table 5). These significant correlations remained after adjusting for changes in TC levels ($p = 5.4 \times 10^{-5}$, $Q = 0.0005$; $p = 0.02$, $Q = 0.08$) or LDL-C levels ($p = 6.8 \times 10^{-5}$, $Q = 0.0006$; $p = 0.02$, $Q = 0.098$). The significant correlation between changes in plasma LPS concentrations and changes in HDL-C levels disappeared after adjusting for changes in TG levels ($p = 0.13$, $Q = 0.4$). In contrast, a significant correlation between changes in plasma LPS concentrations and changes in TG levels was independent of changes in HDL-C levels ($p = 0.0003$, $Q = 0.002$). The statistical analysis shown in Tables 3–5 was also performed on the dataset excluding the data of type 1 diabetes (T1D) patients, but the results were similar to those including T1D (Tables S2–S4). The correlation between plasma LPS concentrations and TGs in patients with AGM, shown in Tables 3–5, is illustrated in Figure 5.

**Table 5.** Multiple regression analysis between changes in physiological and biochemical test values and changes in plasma LPS concentrations.

| Characteristic | Model 1 | | | Model 2 | | |
|---|---|---|---|---|---|---|
| | β | *p* | *Q* | β | *p* | *Q* |
| FPG | 5 | 0.99 | 1.00 | 69 | 0.91 | 0.99 |
| GA | −27 | 0.53 | 0.79 | −1 | 0.98 | 1.00 |
| TGs | 2041 | <0.01 | <0.05 *** | 2076 | <0.01 | <0.05 *** |
| TC | −78 | 0.83 | 0.97 | −26 | 0.94 | 1.00 |
| LDL-C | −139 | 0.66 | 0.90 | −82 | 0.80 | 0.97 |
| HDL-C | −208 | <0.05 | <0.1 ** | −225 | <0.05 | <0.1 ** |
| AST | 21 | 0.95 | 1.00 | 13 | 0.97 | 1.00 |
| ALT | 104 | 0.81 | 0.97 | 87 | 0.85 | 0.97 |
| γ-GTP | 145 | 0.34 | 0.59 | 145 | 0.36 | 0.60 |
| SBP | 207 | 0.32 | 0.57 | 230 | 0.28 | 0.52 |
| BMI | 12 | 0.23 | 0.46 | - | - | - |
| hs-CRP | 2 | 0.41 | 0.65 | 2 | 0.42 | 0.67 |

LPS, lipopolysaccharide; FPG, fasting plasma glucose; GA, glycated albumin; TGs, triglycerides; TC, total cholesterol; HDL-C, high-density lipoprotein cholesterol; LDL-C, low-density lipoprotein cholesterol; AST, aspartate aminotransferase; ALT, alanine aminotransferase; γ-GTP, γ-glutamyl transferase; SBP, systolic blood pressure; BMI, body mass index; hs-CRP, high-sensitivity C-reactive protein. ** $Q < 0.1$ or *** $Q < 0.05$. Multiple regression analysis was performed. Model 1: unadjusted; Model 2: adjusted for changes in BMI.

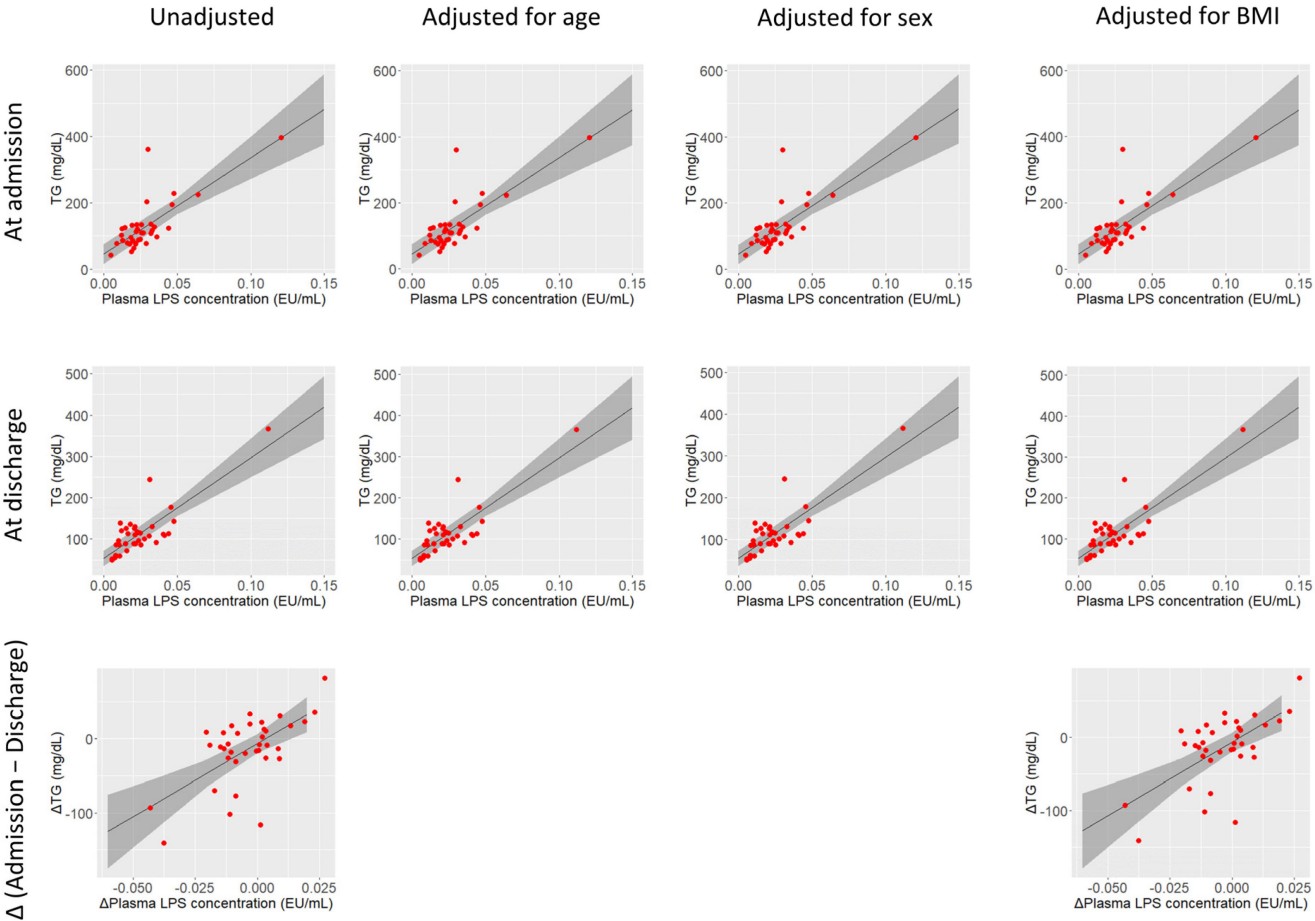

**Figure 5.** Relationships between plasma LPS concentrations and TGs in AGM patients. The correlation between plasma LPS concentrations and TGs in patients with AGM, shown in Tables 3–5, is illustrated in the figures. Red circles represent measured values; solid black lines represent regression lines predicted by the multiple regression analysis; and grey-filled areas represent 95% confidence intervals of the regression lines. LPS, lipopolysaccharide; TG, triglyceride; BMI, body mass index.

## 4. Discussion

In this study, we investigated the relationship between plasma LPS concentrations and health status in healthy subjects and patients with AGM in Japan. The results showed that plasma lipopolysaccharide concentration correlated with obesity (BMI), glucose metabolism (CPR, FPG, and 20/(C-peptide × FPG)), lipid metabolism (TGs and HDL-C), and renal function (stage of diabetic nephropathy).

### 4.1. Characteristics of the Study Subjects

In Japan, population surveys are conducted by the Statistics Bureau of the Ministry of Internal Affairs and Communications. In addition, the National Health and Nutrition Survey (NHNS), a survey of the overall health of the population, is regularly conducted by the Ministry of Health, Labour, and Welfare.

The closest population survey to the time of this study was conducted in 2016 [46]. The results of the population survey showed that the median age was 47 years and the proportion of males was 49%. On the other hand, the median age of the subjects in study 1 was 33 years, and the proportion of male was 67%. The results of the NHNS conducted in 2017 [47], the same year as this study, showed that the mean BMI values were 24 kg/m$^2$ for males and 23 kg/m$^2$ for females, the drinking rate was 20%, and the smoking rate was 18%. On the other hand, the mean BMI values of the subjects in Study 1 were 22 kg/m$^2$ for males and 21 kg/m$^2$ for females; the drinking rate was 69%; and the smoking rate was 11%. From these results, it can be concluded that Study 1 subjects were younger, more likely to be male, and leaner than the Japanese population as a whole. Study 1 subjects were also found to have higher rates of alcohol consumption and, conversely, lower rates of smoking than the Japanese population as a whole. However, as no association was found between drinking/smoking habits and plasma LPS concentrations in this study, differences in drinking and smoking rates between the subjects of Study 1 and the Japanese population as a whole may not affect the interpretation of this study.

In a previously published cross-sectional study of Japanese patients with type 2 diabetes [48,49], the means ± SDs of each demographic and clinical parameter were reported as follows (corresponding data from Study 2 are given in brackets): age: 64.7 ± 12.6 and 62.5 ± 10.8 years (vs. 61.9 ± 15.3 years); proportion of males: 69% and 52% (vs. 58%); CPR: 1.8 ± 1.3 ng/mL (vs. 2.3 ± 1.4 ng/mL); HbA1c: 9.5 ± 2.1% (vs. 9.0 ± 2.6%); FPG: 155.3 ± 44.7 mg/dL (vs. 133.1 ± 44.5 mg/dL); TGs: 124.9 ± 59.1 mg/dL (vs. 127.4 ± 74.4 mg/dL); TC: 190.3 ± 45.5 mg/dL (vs. 178.4 ± 33.3 mg/dL); HDL-C: 46.1 ± 16.8 mg/dL or 46.8 ± 13.9 mg/dL (vs. 43.8 ± 12.1 mg/dL); LDL-C: 106.5 ± 36.8 mg/dL (vs. 105.6 ± 26.8 mg/dL); ALT: 31.2 ± 31.1 U/L (vs. 35.3 ± 42.5 U/L); SBP: 131.7 ± 19.3 mmHg (vs. 129.6 ± 21.7 mmHg); BMI; 25.3 ± 4.7 kg/m$^2$ (vs. 28.5 ± 8.0). Based on the above, it can be assumed that the subjects in Study 2 were not obviously different from the general AGM patients in Japan in terms of their main health status.

### 4.2. Blood LPSs and Obesity

A significant increase in body weight has been reported in mice continuously treated with LPSs subcutaneously for 28 days [1], suggesting that LPSs in the blood may contribute to obesity. Regarding the mechanism, human and in vitro studies have shown that LPSs suppress brown adipose tissue-specific gene expression [50]. It has also been reported that blood levels of cortisol, an appetite-stimulating hormone, increase in humans injected with LPSs [51,52] and that there is a positive correlation between diurnal variations in both blood LPS levels and appetite [25], suggesting that blood LPSs may also lead to obesity in terms of energy intake. Previous reports investigating the association between blood LBP and BMI in the Japanese population have found a positive correlation between blood LBP concentration and BMI in healthy subjects [12]. In the present study, a significant positive correlation was found between plasma LPS concentration and BMI in healthy subjects. On the other hand, demographics (age, sex), lifestyle (smoking, drinking), and dietary habits (number of meals, and frequency of snacks and fatty meals), which have been reported to

be associated with plasma LPS concentration, were not associated with plasma LPS levels in healthy subjects. In other words, it is speculated that plasma LPS concentration may be associated with BMI in healthy subjects independently of these factors. As mentioned above, it has been suggested that LPSs may contribute to the induction of obesity through several mechanisms. In particular, with regard to adipocyte browning, although there are few reports of an association between blood LPS levels and adipocyte browning in healthy subjects, it has been reported that the expression of browning-related genes is reduced when LPSs are applied to primary adipocytes derived from healthy subjects in vitro [50]. It has also been reported that there is a negative correlation between the expression of browning-related genes in adipose tissue and BMI [50]. Therefore, the suppression of adipocyte browning by LPSs may be behind the significant positive correlation between plasma LPS concentration and BMI in healthy subjects in the present study. The detailed mechanism of this correlation needs to be elucidated in further studies.

Previous reports in the Japanese population have found a positive correlation between blood LBP concentration and BMI in type 1 [15] and type 2 [14] diabetics. However, our study found that plasma LPSs correlated with BMI in healthy subjects but not in AGM patients. The BMI range (minimum–maximum) of the healthy subjects in this study was 18–29 kg/m$^2$, while the BMI ranges of the AGM group were 17–49 kg/m$^2$ before admission and 16–46 kg/m$^2$ after admission; there is sufficient variation in BMI in the AGM group to analyse the correlation between BMI and blood LPS concentration. On the other hand, 94% of the AGM group in this study was on medication, so the association between blood LPS levels and BMI may have gone undetected due to medication-induced obesity. Another possibility is that abnormal lipid metabolism in diabetic patients may have disturbed the correlation between blood LPS concentration and BMI. Although 90% of blood LPSs are bound to lipoproteins, it has been reported that in diabetic patients, reduced VLDL catabolism results in increased LPSs in the VLDL and HDL fractions, or free LPSs, and decreased LPSs in the LDL fraction [53]. The reactivity of LPSs with LAL reagents varies depending on the type of lipoprotein to which they are bound [54]. Therefore, differences in the lipid metabolic capacity of individual diabetic patients may have influenced the measured LPS levels, independent of BMI. Note that the reactivity of LPSs with LAL reagents is related to the LPS-stimulated production of inflammatory cytokines by monocytes [55]. In other words, the effects of LPSs on diabetic patients may be independent of BMI. This may be a phenomenon that has not been captured by previous reports that only measured blood LBP levels in Japanese diabetic patients.

### 4.3. Blood LPSs and Glucose Metabolism

As described in the "Section 1", LPSs have been shown in in vitro studies using myoblast cell lines to increase insulin resistance through inducible nitric oxide synthase-mediated tyrosine nitration of insulin receptor substrate 1 [7]. It has been reported that Japanese patients with type 2 diabetes have higher levels of LBP in their blood than healthy subjects [13]. Epidemiological studies in Japanese patients with type 2 diabetes have also reported that blood LBP concentrations are positively correlated with HbA1c, FBG, and CPR [14]. In our study, even after adjusting for basic characteristics such as age and BMI (which have been reported to be associated with plasma LPS concentrations [31] and which confirmed intergroup differences between healthy subjects and AGM patients in this study), AGM patients had significantly higher plasma LPS concentrations compared with healthy subjects. Furthermore, there was a significant correlation between the plasma LPS concentration and CPR, FPG, and insulin resistance (20/(CPR × FPG)), respectively, in patients with AGM. These results suggest the possibility that there is an association between plasma LPS concentration and impaired glucose metabolism in the Japanese. In contrast, no significant correlation was found between blood LPS levels and HbA1c in the present study. This may be due to the different half-lives of LPSs and LBP in the blood. In a study in which LPSs were injected into human blood, the blood concentration of LPSs peaked after 5 min and disappeared after 15 min [56]. On the other hand, the increase in blood LBP

concentrations induced by LPS injection is maintained 3 days after LPS injection [57]. As HbA1c is a measure of past hyperglycaemic state, it may, by its nature, be more strongly associated with LBP concentrations than with blood LPS concentrations. In Study 2, plasma LPS concentrations did not decrease in patients with AGM, despite significant decreases in FPG and GA levels during hospitalization. In this study, patients were given exercise and dietary advice, which might have improved their glucose metabolism independently of the effect of plasma LPS concentration.

### 4.4. Blood LPSs and Lipid Metabolism

In vitro studies using rabbit hepatocytes have reported that stimulation with LPSs increases the amount of TGs secreted [58]. As mentioned above, LPSs have also been reported to inhibit the browning of adipocytes, which is important for triglyceride metabolism. Furthermore, LPS-induced increases in TG levels have also been reported in the blood of LPS-treated humans [57] and in the livers of mice [1]. In an epidemiological study of Japanese patients with type 2 diabetes, blood LBP levels were negatively correlated with HDL-C but not with TGs [14]. On the other hand, in this study, we found that plasma LPS concentrations were significantly associated with lipid profiles, particularly with TG levels in Japanese AGM patients. It has been reported that 90% of LPSs in human plasma are bound to lipoproteins [53]. Therefore, we questioned whether the high correlation between plasma LPS concentrations and TG levels may be due to in vivo co-localization. The major lipoproteins are chylomicrons, very-low-density lipoproteins (VLDL), LDL, and HDL, with the first two being the most TG-rich lipoproteins. A previous study evaluating the distribution of LPSs in the VLDL, LDL, and HDL fractions in the serum of fasting healthy subjects showed that most LPSs were localized in the VLDL fraction [59]. In contrast, a previous report showed that the distribution of LPSs in the VLDL, LDL, and HDL fractions did not differ, suggesting that LPSs do not necessarily co-localize with TGs in the blood [53]. The other concern was that TGs may cause false-positive results in the LAL reaction because they absorb light at 405–410 nm [60]. In this regard, we confirmed that our method for measuring LPS concentration is not interfered with by triglycerides (Figure S6). Many studies have reported that blood LPS concentrations correlate with TG levels. In an epidemiological study among 6632 people in Finland, there was a positive correlation between the serum LPS concentration and TG levels and a negative correlation between the serum LPS concentration and HDL-C levels [4]. A cross-sectional study conducted in the UK in healthy, obese, glucose-intolerant, and type 2 diabetes subjects also found a positive correlation between the serum LPS concentration and TG levels [61]. Thus, many reports have suggested that LPSs contribute to an increase in TG levels, and we believe that a similar relationship was observed in Japanese patients with AGM.

### 4.5. Blood LPSs and Renal Functions

Previous reports measuring blood LBP levels in Japanese diabetic patients either did not assess renal function [13] or found no correlation between blood LBP levels and renal function [19]. Our study showed a significant positive correlation between the plasma LPS concentration and the stage of diabetic nephropathy in patients with AGM. Diabetic nephropathy is the leading cause of end-stage renal disease, and an inflammatory response is thought to be involved in its development. In a previous study, serum LPS concentrations in patients with diabetes and macroalbuminuria were higher than those in patients with diabetes and microalbuminuria or a normal albumin excretion rate [62]. In addition, LPS administration promoted nephropathy in a diabetic mouse model [63,64]. The study could not identify the mechanisms behind the correlation between plasma LPS concentrations and the stage of diabetic nephropathy, but a triglyceride-mediated mechanism is described in the next section as a possible mechanism.

*4.6. Estimation of the Main Targets of Blood LPSs in AGM Patients*

As described above, in Study 2, plasma LPS concentration was associated with glucose metabolism (insulin resistance), lipid metabolism (TGs and HDL-C), and renal function (diabetic nephropathy). In diabetes mellitus, these factors are thought to interact with each other. In this regard, the correlations among the plasma LPS concentration and $20/(CPR \times FPG)$, HDL-C levels, and the stage of diabetic nephropathy disappeared after adjusting for TG levels, whereas the correlation between the plasma LPS concentration and TG levels was not affected after adjusting for other factors. This suggests the possibility that plasma LPS concentration directly affects TG levels and that the correlation between blood LPS concentration and $20/(CPR \times FPG)$, HDL-C levels, and diabetic nephropathy, respectively, is mediated by TGs. This hypothesis is supported by previous studies as follows: As described above, LPSs are suggested to induce TG synthesis in the liver [1,60]. Ectopic accumulation of TGs in the liver and muscle (outside of adipose tissue) can cause tissue damage and increase insulin resistance in these tissues [65,66]. Elevated TG levels also activate cholesteryl ester transfer protein, which results in the reduction in circulating HDL-C [67]. Furthermore, TG-rich lipoproteins can activate immune responses, disrupt the glomerular filtration barrier, and contribute to the progression of diabetic nephropathy [68]. The findings of these studies, together with our findings, imply the possibility that LPS-induced elevation in TG levels may play a pivotal role in the increase in insulin resistance and decrease in HDL-C levels and renal functions in patients with AGM in Japan. The causal relationships and interactions between plasma LPS concentrations and individual clinical parameters in AGM patients need to be clarified in further studies.

*4.7. Implications for Clinical Applications*

The potential clinical applications of plasma LPS concentrations suggested by the results of this study can be categorised into two main aspects. Firstly, there is the possibility of using these findings in the prevention of obesity. As mentioned in Section 4.1, plasma LPS level has been suggested as a potential factor contributing to obesity. Study 1 showed an association between plasma LPS concentration and BMI in healthy Japanese subjects. These results suggest that LPS-induced obesity may already be occurring in healthy individuals. In Japan, there is a well-established system for the annual health screening of workers and residents. In the future, plasma LPS concentration could potentially be used as an indicator of obesity risk in such health examinations.

Secondly, there is the potential use of plasma LPS concentrations in determining therapeutic strategies for individuals with AGM. As described in Section 4.3, plasma LPS level has been shown to be a factor contributing to elevated blood TG levels. Study 2 showed a strong correlation between plasma LPS concentration and blood TG levels in AGM patients. This suggests that plasma LPSs may contribute to elevated TG levels in AGM patients. TGs are known to represent a risk factor for macrovascular and microvascular complications in diabetic patients, emphasising the importance of managing blood TG levels [69]. Dietary interventions are currently used as one of the approaches to reducing blood TG levels in diabetic patients [69]. Dietary strategies include reducing energy and saturated fat intake while increasing intake of monounsaturated fatty acids, polyunsaturated fatty acids, and dietary fibre [69]. On the other hand, dietary factors such as oligosaccharides, probiotics, and polyphenols have been reported to potentially reduce plasma LPS concentrations [9]. For example, an intervention trial using *Bifidobacterium longum* and oligofructose in patients with non-alcoholic steatohepatitis reported a reduction in blood TG levels associated with a reduction in plasma LPS concentrations [70]. If the role of plasma LPSs in elevating blood TG levels in AGM patients is further demonstrated, it may be possible, in the future, to individualise dietary interventions based on each patient's plasma LPS concentration, allowing for more effective and personalised interventions.

*4.8. Limitations*

There are several limitations in the present study. First, this study involved residents of the prefectures of Tochigi and Miyagi in Japan. As described in Section 4.1, the subjects in Study 1 differed from the Japanese population in terms of age, sex, and BMI distribution. Therefore, the results of this study cannot be generalised to the Japanese population as a whole; thus, further research is needed to determine whether these results can be extrapolated to other regions in Japan.

Second, dietary habits, exercise habits, or the presence of comorbidities may be confounding factors in the results of this study. In Study 1, we assessed energy and lipid intake, which is known to affect blood LPS concentrations [32], with a simple questionnaire and found that in the subjects, they were not associated with LPS concentrations. In addition, although no dietary surveys were conducted on the study subjects in Study 2, their diets during their hospitalisation were managed according to Japan Diabetes Society guidelines. Specifically, their diet was controlled to be 30 kcal × ideal body weight, as described in Section 2.4.2. Therefore, we consider that in Study 2, the influence of energy and lipid intake on the study results was minimised. With regard to comorbidities, there was a concern that gastrointestinal diseases and acute infections might have affected plasma LPS concentrations. However, Study 1 found no association between these factors and plasma LPS concentrations, and in Study 2, those with these complications were excluded from the study population. On the other hand, the study did not examine the intake of detailed dietary factors, nor did it obtain data on physical activity and the presence of complications other than gastrointestinal diseases and acute infections, and the possibility that these factors may have influenced the results has not been excluded.

Third, Study 1 (healthy subjects) and Study 2 (AGM) were conducted in different regions, time periods, and institutions. Therefore, when conducting such studies, there are concerns about potential differences in participants' lifestyles, and dietary and exercise habits, as well as differences in the methods for analysing the samples. Among these factors, it cannot be excluded that the observed between-group differences in plasma LPS concentrations could have been due to differences in lifestyle, diet, and exercise habits, as comparable data on these aspects were not obtained in Studies 1 and 2. With regard to the analytical methods, in particular the analysis of plasma LPS concentrations, the same reagents and protocols were used in Studies 1 and 2 to minimise inter-institutional differences, thus eliminating this potential source of variation.

Fourth, the causality of the correlations obtained here cannot be inferred because the present study is a cross-sectional study. A statistical analysis method called propensity score matching is used to infer causality from the results of cross-sectional studies [71]. However, when using propensity score matching, it is crucial to obtain a wide range of potentially confounding factors that may influence the test values of interest. In this study, basic characteristics reported to influence blood LPS concentrations, such as age, sex, and BMI, were collected. However, detailed data on lifestyle habits, in particular diet and exercise habits, were not collected in this study. Therefore, given the study design of the present study, there were insufficient data to make causal inferences. We consider it necessary to establish a causal relationship among the observed associations between plasma LPS concentrations and clinical parameters in this study by collecting a wide range of data on potential confounders and by conducting a prospective cohort study to establish causal relationships among these items.

Fifth, the present study set the sample size based on clinical parameters that have been reported to show a strong correlation with blood LPS concentration. Therefore, there is a concern that false negatives may occur for clinical parameters with a weak correlation with blood LPS concentration. Specifically, in Study 1, no correlation was observed between age [31], sex [31], smoking [32], and alcohol habits [31]—previously reported to be associated with blood LPS concentration—and plasma LPS concentration, respectively. Additionally, in Study 2, despite the reported association between blood LPS concentration and HbA1c [31], LDL-C [31], SBP [31], BMI [31], and hs-CRP [72], respectively,

in individuals with type 2 diabetes, no correlation was found in this study. These results do not conclusively establish the absence of correlation for these factors, warranting further investigations with higher statistical power to re-evaluate the relationships.

Sixth, an FDR of 0.2 was used to correct for *p*-values in this study. Therefore, it is important to note the potential for false positives in items with larger *p*-values among the clinical parameters that showed a correlation with plasma LPS concentration in this study. For these reasons, this study is preliminary in nature, and these findings should be verified with longitudinal studies.

## 5. Conclusions

In the present study, plasma LPS concentrations were positively correlated with BMI in healthy subjects. In patients with AGM, plasma LPS levels showed a particularly strong positive correlation with TGs. These results suggest a possible role of LPS influx into the blood in obesity in healthy subjects and in the deterioration of triglyceride metabolism in AGM patients in the Japanese population. Although this is a small-scale, preliminary cross-sectional study, we believe that this study reports important findings, especially considering the limited number of previous studies that have evaluated the relationship between plasma LPS concentration and health status in Japan.

**Supplementary Materials:** The following supporting information can be downloaded at: https://www.mdpi.com/article/10.3390/j6040040/s1, Figure S1: Association between plasma LPS concentrations and number of meals per day, Figure S2: Association between plasma LPS concentrations and snacking frequency, Figure S3: Association between plasma LPS concentrations and frequency of fatty meals, Figure S4: Association between plasma LPS concentrations and the presence or absence of a history of gastrointestinal diseases, Figure S5: Comparison of C-peptide immunoreactivity (CPR) values between insulin users and non-users, Figure S6: Effect of triglycerides on lipopolysaccharide (LPS) measurement, Table S1: *p*-Values from the Shapiro–Wilk test, Table S2: Multiple regression analysis between physiological and biochemical test values and the plasma LPS concentration of patients at admission (T1D excluded), Table S3: Multiple regression analysis between physiological and biochemical test values and the plasma LPS concentration of patients at discharge (T1D excluded), Table S4: Multiple regression analysis between changes in physiological and biochemical test values and changes in plasma LPS concentrations (T1D excluded).

**Author Contributions:** Conceptualization, K.Y.I., Y.U., T.M., H.K., H.U. and H.S.; methodology, N.F., S.S., T.I.-S. and Y.U.; formal analysis, N.F., S.S., T.I.-S. and Y.U.; investigation, N.F., S.S., T.I.-S., Y.U. and I.S.; data curation, N.F., S.S., T.I.-S. and Y.U.; writing—original draft preparation, N.F., S.S. and T.I.-S.; writing—review and editing, K.Y.I., Y.U., I.S., T.M., H.K., H.U. and H.S.; supervision, T.M., H.K., H.U. and H.S.; project administration, K.Y.I. and Y.U.; funding acquisition, K.Y.I. and T.M. All authors have read and agreed to the published version of the manuscript.

**Funding:** Study 2 was supported by the Centre of Innovation Program of the Japan Science and Technology Agency (JST) under Grant No. JPMJCE1303.

**Institutional Review Board Statement:** This study was conducted in accordance with the guidelines of the 2013 version of the Declaration of Helsinki, and all procedures involving human subjects were ap-proved by the ethics committees of KAGOME CO., LTD. (approval Nos. 2017-R03 for Study 1 and 2017-R13 for Study 2), and the ethics committees of Tohoku University (approval No. 2018-1-061 for Study 2).

**Informed Consent Statement:** Informed consent was obtained from all subjects involved in the study.

**Data Availability Statement:** The data presented in this study are available on request from the corresponding author. The data are not publicly available due to ethical concerns.

**Acknowledgments:** We thank all the participants and the staff of NIGHTINGALE Inc. for the blood collection.

**Conflicts of Interest:** Studies 1 and 2 were self-funded by KAGOME CO., LTD. KAGOME CO., LTD., engages in health-promoting business, and N.F., Y.U., H.U. and H.S. are employees of the company.

In addition, N.F., Y.U. and H.U. hold stocks of KAGOME CO., LTD. The paper reflects the views of the scientists, not the company's.

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
