# Peer review of "Relationship between Plasma Lipopolysaccharide Concentration and Health Status in Healthy Subjects and Patients with Abnormal Glucose Metabolism in Japan: A Preliminary Cross-Sectional Study"

_2571-8800, doi:10.3390/j6040040_

Round 1
Reviewer 1 Report
Comments and Suggestions for Authors
1. The sample size for the two cross-sectional studies was calculated based on data from previously published studies. So, should the study results be considered preliminary or hypothesis-generating?
2. Given that the two studies were not adequately powered, comment on false positives in the study results. The authors did not adjust the p-value for the multiplicity of comparisons. Justify.
3. Based on my above comments, the results of this study cannot be taken as conclusive, unless the authors have a strong justification to do so.
4. The study limitations should include the small sample size and the possibility of false positives.
5. Discuss the implications of the study findings.
Reviewer 2 Report
Comments and Suggestions for Authors
- What are the limitations of this cross-sectional studies? How do address these limitations in this study?
- authors repoered that plasma lipopolysaccharide concentration was higher in patients with abnormal glucose metabolism than in healthy subjects. However, they did not control for other potential confounding factors, such as diet, exercise, and smoking. Is it possible that these factors could explain the difference in plasma lipopolysaccharide concentration between the two groups?
- The plasma lipopolysaccharide concentration was positively correlated with C-peptide, fasting plasma glucose levels, triglyceride, and stage of diabetic nephropathy, and negatively correlated with 20/(C-peptide × fasting plasma glucose) and high-density lipoprotein cholesterol. These correlations are suggestive of a link between plasma lipopolysaccharide concentration and metabolic syndrome. However, the authors did not perform any causal inference analyses. Can the authors make any causal inferences from their data?
- What are the clinical implications of the study findings?
How did the authors measure plasma lipopolysaccharide concentration?
- What were the demographics of the study participants?
- Were there any differences between the two study groups in terms of age, sex, or other demographic characteristics?
- What statistical tests did the authors use to analyze their data?
- What were the results of the moderator analyses?
- What are the authors' hypotheses about the mechanism by which plasma lipopolysaccharide concentration may be linked to metabolic syndrome?
Reviewer 3 Report
Comments and Suggestions for Authors
Comments to the authors:
In this study, the authors explored the Relationship between plasma lipopolysaccharide concentration and health status in healthy subjects and patients with abnormal glucose metabolism in Japan. The authors provide some interesting data, however, some concerns need to be addressed by the authors before publication. The specific comments see below:
1: Limited Contextual Background: The introduction lacks context on why lipopolysaccharide (LPS) levels are relevant to health status. A brief overview of the physiological implications of LPS and its association with health issues would enhance the reader's understanding.
2: Sample Size Justification: The justification for sample sizes in Study 1 and Study 2 is mentioned but lacks a detailed statistical power analysis. A more comprehensive explanation of the rationale behind the chosen sample sizes would strengthen the study's methodology.
3: Study Design and Controls: The study design is described in detail, but there is no mention of potential confounding factors or efforts to control them. Factors such as diet, lifestyle, and comorbidities could significantly impact the results. A discussion on how these variables were addressed or controlled for is crucial.
4: Ethical Considerations: Although ethical approval is mentioned, there is no discussion about potential ethical concerns related to the study, especially considering the involvement of a private company in the research. It would be valuable to elaborate on any potential conflicts of interest and how they were managed.
5: Correlation vs. Causation: The study discusses correlations between LPS levels and various health parameters. However, it's essential to highlight that correlation does not imply causation. A clear distinction between correlation and causation should be made to avoid misinterpretation of the findings.
6: Geographical Generalization: The study claims a lack of information on LPS levels in Asia, particularly Japan. However, it would be prudent to avoid making broad generalizations about an entire continent based on a study with a relatively small sample size from specific regions in Japan.
7: Insufficient Discussion on Results: The results section is presented, but there is limited discussion or interpretation of the findings. A more thorough discussion of the implications of the results in the context of existing literature and potential clinical applications is needed.
8: Statistical Methods: The statistical methods section is detailed, but the use of specific statistical tests should be justified more explicitly. Additionally, corrections for multiple comparisons should be considered, especially when analyzing multiple correlations.
9: Data Presentation: The data presentation is extensive, but visual aids such as figures or tables could enhance the reader's comprehension. Providing visual representations of key correlations or trends would make the results more accessible.
10: Incomplete Conclusion: The conclusion section is incomplete and does not summarize the key findings or their broader implications. A more comprehensive conclusion should be provided, summarizing the main outcomes and suggesting potential avenues for future research.
11: Timeliness of Information: The study's information is up to December 2018. Given the rapidly evolving nature of scientific research, a brief discussion on any developments or studies post-2018 related to LPS and health status in Japan would be beneficial.
Comments on the Quality of English LanguageModerate editing of English language required
Round 2
Reviewer 2 Report
Comments and Suggestions for Authors
Accept in current form
Reviewer 3 Report
Comments and Suggestions for Authors
The authors have addressed all the concerns. I do not have further comments.